

# Impact of Spatial Distribution Information of Rainfall in Runoff Simulation Using Deep-Learning Methods

Yang Wang[1], Hassan A. Karimi[1]

[1]Geoinformatics Laboratory, School of Computing and Information, University of Pittsburgh, 135 N Bellefield Ave, Pittsburgh, PA 15213, USA

*Correspondence to*: Yang Wang (yaw70@pitt.edu)

**Abstract.** Rainfall-runoff modelling is of great importance for flood forecast and water management. Hydrological modelling is the traditional and commonly used approach for rainfall-runoff modelling. In recent years, with the development of artificial

intelligence technology, deep learning models, such as the long short-term memory (LSTM) model, are increasingly applied to rainfall-runoff modelling. However, current works do not consider the effect of rainfall spatial distribution information on the results, and the same look-back window is applied to all the inputs. Focusing on two catchments from the CAMELS dataset, this study first analyzed and compared the effects of basin mean rainfall and spatially distributed rainfall data on the LSTM models under different look-back windows (7, 15, 30, 180, 365 days). Then the LSTM+1D CNN model was proposed to

simulate the situation of short-term look-back windows (3, 10 days) for rainfall combined with the long-term look-back windows (30, 180, 365 days) for other input features. The models were evaluated using the Nash Sutcliffe efficiency coefficient, root mean square error, and error of peak discharge. The results demonstrate the great potential of deep learning models for rainfall runoff simulation. Adding the spatial distribution information of rainfall can improve the simulation results of the LSTM models, and this improvement is more evident under the condition of short look-back windows. The results of

the proposed LSTM+1D CNN are comparable to those of the LSTM model driven by basin mean rainfall data and slightly worse than those of spatially distributed rainfall data for corresponding look-back windows. The proposed LSTM+1D CNN provides new insights for runoff simulation by combining short-term spatial distributed rainfall data with long-term runoff data, especially for catchments where long-term rainfall records are absent.





## 1 Introduction

Rainfall-runoff simulations are vital for watershed water resources management and risk analysis (Montanari, 2005; Neitsch et al., 2011). In addition, rainfall-runoff simulation plays an increasingly important role as a technical basis for hydrological forecasting due to the frequent occurrence of extreme hydrological events caused by climate change(Grayman, 2011; Panagoulia and Dimou, 1997). As the most widespread and essential tool for water science research, hydrological model plays

a pivotal role in the rainfall-runoff simulation (Krause et al., 2005; Sood and Smakhtin, 2015). The development of hydrological models cannot be separated from the continuous research on hydrological processes. It is on the basis of the continuous understanding of hydrological processes that hydrological researchers have enough theoretical basis for building models that describe the interrelationship between the various hydrological elements and can simulate the overall hydrological cycle. The development of hydrological models has gone through two main stages, namely, lumped hydrological models and

distributed hydrological models (Devia et al., 2015). For example, the Stanford model is the first lumped hydrological model with a solid theoretical basis (CRAWFORD and H., 1966). In 1977, British, Danish and French researchers jointly proposed the SHE hydrological model, which is the first generation of distributed hydrological models (Sahoo et al., 2006). The Variable Infiltration Capacity (VIC) is a large-scale distributed hydrological model developed by the University of Washington, the University of California at Berkeley, and Princeton University (Liang et al., 1996). The distributed VIC model is based on the

idea of gridding to achieve distributed simulation of watersheds.

However, the fact that we cannot accurately describe every process of the hydrologic cycle leads to the necessary simplifications in the hydrologic model calculation process, which is one of the contributing factors to simulation errors. Since models based on physical mechanisms cannot fully describe the physical processes of the hydrologic cycle, researchers started to explore data-driven models for hydrologic modelling (Solomatine and Ostfeld, 2008). For example, Support Vector

Machines (SVMs) are often used to manage the processing of hydrological model input data or to perform hydrological simulations directly due to their advantages in processing nonlinear problems (Ahmad et al., 2010; Sivapragasam et al., 2001). Artificial neural networks (ANNs) are a type of machine learning method that have been used for hydrological modelling since the 1990s. In the following years, more research has demonstrated that ANN models can achieve comparable results to physical models while requiring less data (Chang et al., 2015; Ömer Faruk, 2010). Although the robustness of ANN models needs to

be further investigated, the ability of ANNs to capture the nonlinearity associated with hydrologic applications has led to its widespread use (Ghumman et al., 2011).

In recent years, due to the development of deep learning techniques, such as CNN in the field of image recognition and LSTM in natural language processing and time series data, these techniques have also been widely used in simulation of rainfall-runoff. Among these, LSTM has garnered more attention of researchers due to its suitability for processing and predicting

events with very long intervals and delays in time series. For example, (Hu et al., 2018) compared the difference between ANN and LSTM in simulation of flood events, and the results show that LSTM models perform significantly better than ANN models. (Kratzert et al., 2018) trained LSTM models with rainfall-runoff data from several watersheds, demonstrating the





potential of LSTM as a regional hydrological model, one of which can predict flows in various watersheds. A LSTM model was also used in combination with Sequence-to-Sequence to simulate the discharge for the next few hours (Xiang et al., 2020).

(Gauch et al., 2020) 's study illustrated that LSTM can process different input variables at different time scales. CNNs are another deep learning approach that have received increasing attention in recent years (Shen, 2018) . CNNs are often used to process data with spatial distribution information, such as images. Some studies have combined CNN and LSTM in the hope of better processing spatio-temporal data mining, such as rainfall-runoff simulation.(Wang et al., 2019). For example, (Liu et al., 2021) first used CNN to process the input data at each time step, and then fed the resulting vectors into LSTM, and the

results showed that the model performed better in simulating the peak flood.

In summary, current works have the following shortcomings. Firstly, the deep learning model represented by LSTM for rainfall-runoff simulation does not focus on the spatial distribution information of rainfall. It is known that rainfall is the most direct and influential factor on the formation of runoff. Current research mostly uses surface-mean rainfall to drive LSTM models, which to some extent loses the spatial distribution information of rainfall, which in a physical sense has a very

important impact on the formation of runoff, especially the formation of peak discharge.

Second, the length of sequence of different input data for hydrological modelling by LSTM is consistent. The advantage of LSTM is that it can perform better in longer sequences than a normal Recurrent Neural Network (RNN). However, if this problem is looked at from the point of view of the physical mechanism of rainfall-runoff formation, for example, only the rainfall that occurred in the previous few days usually has an impact on the current moment of discharge. Rainfall that occurred

many days prior may not have an impact on the current runoff, and this length of time varies with the characteristics of the watershed. If we use LSTM to process a long series of discharge data, combined with rainfall data from the previous few days, we may get a better simulation result.

Third, there are different types of LSTM models such as 'many to one', which is to predict the value of the next time step using the data of the past $n$ time steps and 'many to many', which is to predict the value of the future multiple steps using the

data of the past $n$ time steps. The current research on rainfall-runoff simulation by LSTM mainly uses the 'many to one' type. Analysing the performances of 'many to many' type of LSTM can help better apply the LSTM model to rainfall-runoff simulation

The main objective of this study is to first explore the difference between the results obtained using a LSTM model driven by rainfall data with spatial distribution information and a LSTM model driven by basin mean rainfall data under different look-

back windows. We focus not only on the simulation of the next one-time step, but also on the simulation of multiple future time steps. To meet this objective, we propose a LSTM+1D CNN model to simulate rainfall-runoff by combining meteorological and discharge data of long look-back window and rainfall data with spatial distribution of short look-back window, and compare the results with the traditional LSTM model.

The paper is structured is as follows. Section 2 describes the data, the proposed LSTM+1DCNN model, and the experimental

design. Section 3 analyses and discusses results. Section 4 provides concluding remarks and discusses future research.



## 2 Methods and Dataset

### 2.1 The CAMELS Dataset

In this study, we use the CAMELS HYDROMETEOROLOGICAL TIME SERIES from the National Center for Atmospheric Research (NCAR) (Addor et al., 2017; Newman et al., 2015). The dataset contains lumped meteorological forcing data and observed flows on a daily time scale starting in 1980 for most basins. Lumped meteorological forcing data were mainly calculated from three grid data sources, namely Daymet (Thornton et al., 2014), Maurer (Livneh et al., 2013), and NLDAS (Xia et al., 2012). We used the Daymet data in this study since it has a resolution of 1 km, which is better than the other two. CAMELS contains a total of 671 catchments with minimal anthropogenic disturbance in the contiguous United States (CONUS). In this study we selected two of these catchments and the hydrologic response units they contain. The basin ID for Catchment 1 is 03164000 (NEW RIVER NEAR GALAX, VA) and the ID for Catchment 2 is 13340000 (CLEARWATER RIVER AT OROFINO, ID). For each catchment, CAMELS has the basin mean forcing (lump) dataset, which includes the driving data when using the lumped hydrologic model. These are: (i) daily cumulative rainfall, (ii) daily minimum air temperature, (iii) daily maximum air temperature, (iv) mean short-wave radiation, and (v) vapor pressure. Here the daily cumulative rainfall is treated as the basin mean rainfall data without spatial distribution information. For each catchment, CAMELS also includes the hydrologic response units it contains. In this study, Catchment 1 had a total of 64 hydrologic response units, while Catchment 2 had a total of 194 hydrologic response units. We extracted the rainfall data of each hydrologic response unit and created a dataset for the corresponding catchment, and regarded it as rainfall with spatial distribution information. For Catchment 1, we used a vector of size 64 to represent the rainfall data with spatial distribution information, and for Catchment 2, the vector has a size of 194. The locations of the two basins are shown in Fig. 1.

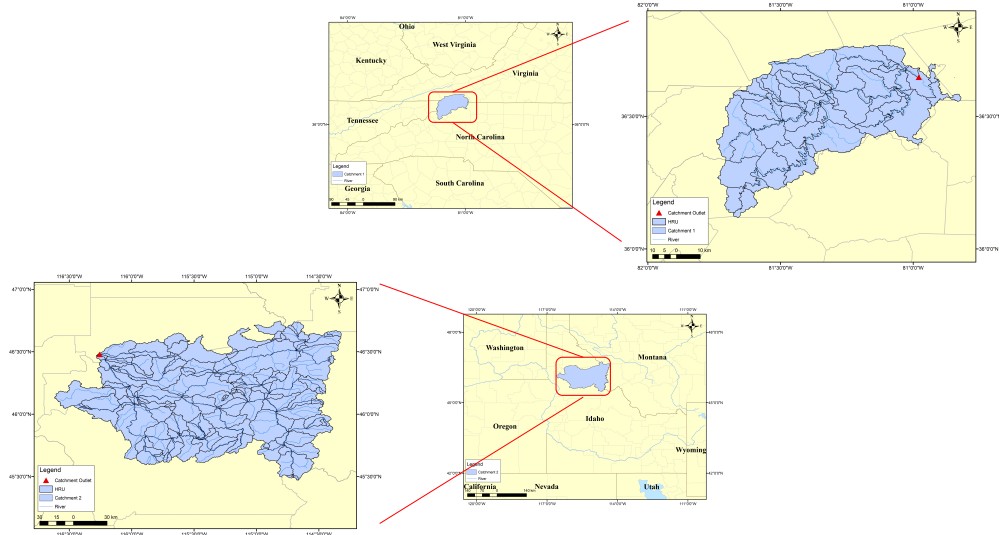

Figure1. Two catchments and their locations in the State


In addition, CAMELS data include simulation results from the hydrologic model, which is the Snow-17 models coupled with the Sacramento Soil Moisture Accounting Model. For each catchment, 10 models were calibrated by root mean squared error

(RMSE) as the objective function, and the model with the lowest RMSE was selected for validation. In this study we also compared the results of different deep learning models with the results of a hydrological model.

## 2.2 Long-short term memory network

RNN is one of the most frequently used models when using deep learning to deal with temporal problems, and the reason why RNN performs well on temporal data is that the input to RNN is the hidden node at time $t-1$ as the current time and the

previous information at time $t$. The main problem with RNN models is the occurrence of long-term dependencies, which arises when the nodes of a neural network have gone through many time steps of computation and the features from a relatively long time ago have been covered by the latest features (Sherstinsky, 2020). The motivation for a LSTM model is to solve the problem mentioned above. As the name implies, Long Short Term Memory is a neural network with the ability to remember both long and short-term information. LSTM was first proposed by (Hochreiter and Schmidhuber, 1997) in 1997, and due to

the rise of deep learning in 2012, LSTM has gone through several generations, resulting in a more systematic and complete LSTM framework that has been widely used in many fields. The reason why LSTM can solve the long-term dependency problem of RNN is that LSTM introduces the gate mechanism for controlling the delivery and loss of features. The basic structure of LSTM is shown in Fig. 2.

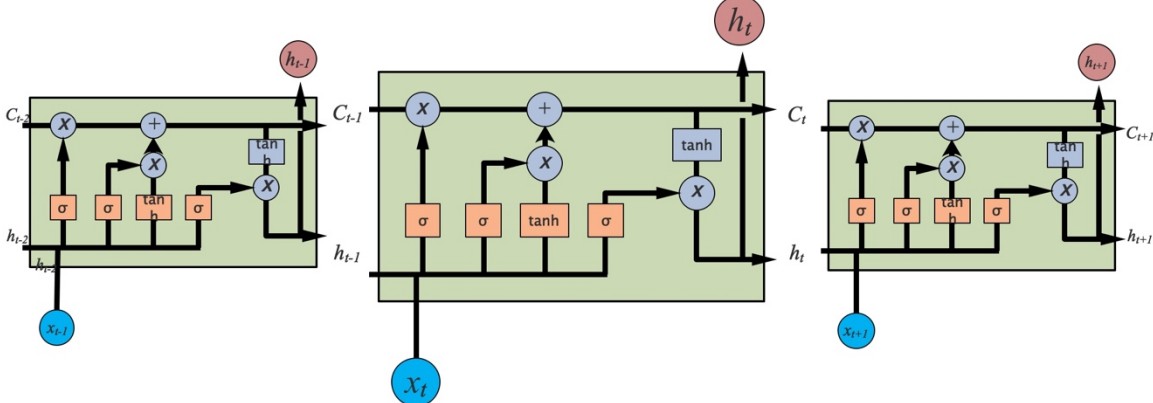

135         Figure 2. Basic LSTM layer structure with a detailed calculation illustration shown in the LSTM cell at time step $t$

Whenever a flow passes through a LSTM cell, there are actions that determine what old information is discarded and what new information is added. The structure that controls the addition and subtraction of information to and from the cell state is called gates. There are three such gates in a LSTM cell, namely forget gate, input gate, and output gate.

The forget gate determines which information needs to be noted and which can be ignored. The information from the current input $x_t$ and the hidden state $h_{t-1}$ is passed through the sigmoid function. Sigmoid generates a value between 0 and 1, which





can be used to describe whether a part of the old output is necessary (by bringing the output closer to 1). This value of f(t) is the output of forget gate.

$$f_t = \sigma \cdot \left( W_f \cdot [h_{t-1}, x_t] + b_f \right) \tag{1}$$

The input gate performs two steps to update the cell state. First, the current state $x_t$ and the previously hidden state $h_{t-1}$ are
passed to a second sigmoid function. Next, the same information about the hidden state and the current state is passed through the tanh function. To regulate the network, the tanh operator creates a vector $c_t$ where all possible values are between -1 and 1.

$$i_t = \sigma \cdot (W_i \cdot [h_{t-1}, x_t] + b_i) \tag{2}$$

$$\overline{c_t} = tanh \cdot (W_c \cdot [h_{t-1}, x_t] + b_c) \tag{3}$$

The next step is to decide and store the information from the new state in the cell state $c_t$. The previous cell state $c_{t-1}$ is multiplied by the forget vector $f_t$. If the result is 0, the information is removed from the cell state. Next, the network takes
the output value of the input vector $i_t$, which updates the cell state and thus provides the network with a new cell state $c_t$.

$$c_t = c_{t-1} \odot f_t + \overline{c_t} \odot i_t \tag{4}$$

The output gate will determine the value of the next hidden state, which contains information about the previous input. First, the model passes the current state and the value of the previous hidden state to a third sigmoid function. The resulting new cell state is then passed through the tanh function. Based on this output value, the network decides what information the hidden state should have. This hidden state is used for output. The new cell state and the new hidden state are transferred to the next
time step.

$$o_t = \sigma \cdot (W_o \cdot [h_{t-1}, x_t] + b_o) \tag{5}$$

$$h_t = o_t \odot \tanh(c_t) \tag{6}$$

In summary, the forget gate determines what relevant information from previous steps is needed. The input gate determines what relevant information can be added to the current step, and the output gate ultimately determines the next hidden state.

In the formula above, $W$s are the weight vectors for different gates ($W_f$ for forget gate, $W_i$ for input gate, $W_c$ for output gate,
and $W_o$ for gate unit). $b$s are the bias vectors for different gates ($b_f$ for forget gate, $b_i$ for input gate, $b_c$ for output gate, and $b_o$ for gate unit). $tanh$ is hyperbolic tangent activation function, and $\sigma$ is sigmoid activation function.

**2.3 Hybrid Model (LSTM + 1D CNN)**

As described above, we aim to process long look-back windows of meteorological data (excluding rainfall) and discharge by LSTM, while we intend to consider the spatial distribution information of rainfall with shorter look-back windows, and finally
combine spatial feature and temporal feature to realize rainfall runoff simulation. The general idea can be expressed as follow:

$$D_t = f(M_t, \ldots, M_{t-n}, D_{t-1}, \ldots, D_{t-n}, P_t, \ldots P_{t-i}) \tag{7}$$

Where $M$ is meteorological data, $D$ is discharge data, and $P$ is rainfall. $n$ is the look-back window for meteorological data, and $i$ is the look-back window for rainfall data.





An LSTM + 1D CNN consists of a 1D CNN and one LSTM. For example, we expect to obtain the simulation results for the discharge at time $t = n$, and our look-back window is 30. We use the traditional LSTM for meteorological data (excluding rainfall) and discharge, which means that at each time step for the look-back window, the meteorological data (excluding rainfall) and discharge at that time are used as input for LSTM. The output of LSTM can be expressed as:

$$H(n) = LSTM(M_n, ..., M_{n-30}, D_{n-1}, ..., D_{n-30}) \tag{8}$$

Rainfall has the most direct effect on runoff generation. In the model we treat past rainfall separately from the current day rainfall. 2D CNN is the most ideal deep learning model to handle spatially distributed data. However, since the input rainfall data in this study come from stream gauge in the watershed, their distribution is irregular. For this, in this study we use 1D CNN to process the vector representing the spatial distribution of rainfall. Suppose we want to consider the rainfall in the past 3 days, for 1D CNN the input size is $(3, M)$, Where $M$ is the number of stream gauges in the watershed and also the length of the vector used to describe the spatial distribution of rainfall. The number of input channels is equal to the number of past days we are considering. Generally, a CNN architecture consists of layers, including convolutional layers, pooling layers, and the activate function. The purpose of the convolutional manipulation is to extract different features of the input layer. As the layer gets deeper, the extracted feature also goes from low-level to high-level. Pool layers use appropriate methods to reduce the number of units input to the next layer, thereby solving the overfitting issue of the model and improving computational efficiency. The output of 1DCNN can be expressed as:

$$S = 1D\ CNN(P_{n-1}, P_{n-2}, D_{n-3}) \tag{9}$$

We also focus on the rainfall on the day $t = n$, and we combine the rainfall at time $n$ $(P_n)$, $H(n)$ and $S$ into a vector $Z_f$ before the final output. At last, the fusion vector $Z_f$ is transformed to the final simulation result, i.e., the estimated discharge value for the basin at time $n$ given the look-back window 30 for meteorological data (excluding rainfall) and discharge, and the look-back window 3 for rainfall data. The general framework is shown in Fig. 3.

$$Z_f = [P_n, H(n), S] \tag{10}$$

$$y_t = w \cdot Z_f + b \tag{11}$$

where $w$ is the neural weight vector for the last layer, and $b$ is the bias.

The same structure can also be used to simulate future multi-day discharges. Since we already have obtained the vector S representing the rainfall information of the past 3 days by 1D CNN, the output of each future day is first combined with $S$, the rainfall of the day, and the LSTM output of the day into a vector, and then the final discharge is obtained.







Figure 3. Framework of proposed LSTM + 1DCNN model (Upper: one time step output with 30 days look-back window for meteorological data (excluding rainfall) and discharge, 3 days look-back window for rainfall; bottom: three time steps output with 30 days look-back window for meteorological data (excluding rainfall) and discharge, 3 days look-back window for rainfall.)

## 2.4 Performance Evaluation Criteria

In this study, the performance of each model is evaluated by statistical error measurements and characteristics of discharge process error including Nash-Sutcliffe efficiency coefficient and root mean square error.

The Nash-Sutcliffe efficiency coefficient (NSE) is generally used to verify the goodness of the hydrological model simulation results. NSE is calculated as follows:



$$NSE = 1 - \frac{\sum_1^n (f^t - q^t)^2}{\sum_1^n (q^t - \overline{q^t})^2} \qquad (12)$$

where $f^t$ is the model simulation discharge at time $t$, $q^t$ is the observed discharge at time $t$, and $\overline{q^t}$ is the mean of observed discharge. NSE takes the value of negative infinity to 1. NSE close to 1 means that the model quality is good and credible; NSE close to 0 means that the simulation results are close to the mean level of the observed values, i.e., the overall results are credible, but the simulation error is large; if NSE is much less than 0, the model is not credible.

The RMSE assesses how well the predictions match the observations. Depending on the relative range of the data, values can range from 0 (perfect fit) to $+\infty$ (no fit). RMSE is calculated as follows:

$$RMSE = \sqrt{\frac{\sum_1^n (f^t - q^t)^2}{n}} \qquad (13)$$

where $f^t$ is the model's simulation discharge at time $t$, $q^t$ is the observed discharge at time $t$. $n$ is the length of the sequence.

We also used the error of peak discharge (EPD) to measure the ability of the model to simulate peak discharge. Since there are multiple peak discharges in the sequence, we use the mean of all peak discharge EPDs as an indicator. EPD can be calculated as follows:

$$EPD = \frac{1}{n} \cdot \sum_1^n \frac{f^t{}_p - q^t{}_p}{q^t{}_p} \cdot 100\% \qquad (14)$$

where $q^t{}_p$ is the observed peak discharge at time $t$, $f^t{}_p$ is the modelled peak discharge at time $t$. $n$ is the number of peak discharges in the dataset.

**2.5 Experimental Setup**

Considering the start and end times of rainfall data for all stations in the two catchments, training data for all models are from January 1, 1980 to December 31, 2008; calibration data for all models are from January 1, 2009 to June 4, 2010; all models were evaluated using data from June 6, 2010 to December 23, 2011. We use $M$ for meteorological data including daily minimum air temperature, daily maximum air temperature, mean short-wave radiation, and vapor pressure; $D$ for discharge data; and $P$ for rainfall data. We trained and tested four experiments (Experiments 1-4) as shown in Table 1.

Table 1. Input data, output, and model selection for four experiments

| ID | Input data before time $t$ | Input data at time $t$ or $t+n$ | Model | Input rainfall | Type of rainfall | Output |
|---|---|---|---|---|---|---|
| Exp. 1 | $M, D, P$ | $M, P$ | LSTM | - | 1. basin mean rainfall data<br>2. spatially distributed rainfall | Discharge for the next one day ($t$) |



| Exp. 2 | $M, D, P$ | $M, P$ | LSTM | - | 1.basin mean rainfall data<br>2.spatially distributed rainfall | Discharge for the next few days $(t + n)$ |
| Exp. 3 | $M, D$ | $M$ | LSTM+<br>1D CNN | Past $i$ days | 2.spatially distributed rainfall | Discharge for the next day $(t)$ |
| Exp. 4 | $M, D$ | $M$ | LSTM+<br>1D CNN | Past $i$ days | 2.spatially distributed rainfall | Discharge for the next few days $(t + n)$ |

Experiment 1 and Experiment 3 are 'one time step output' simulations, which means that these two experiments are used to

simulate the next day discharges. In Experiment 1, we consider 5 different look- back windows, 7 days, 15 days, 30 days, 180 days, and 365 days. We used basin mean rainfall data and spatially distributed rainfall driven LSTM models, respectively, with the aim of investigating whether rainfall data with spatial distribution information would improve the simulation accuracy of the model.  In Experiment 3, we consider 2 different look-back windows for rainfall, which are 10 days and 3days, and 3 different look-back windows for other input, which are 30 days, 180 days and 365 days. We used the proposed LSTM+1D

CNN model to investigate whether processing short-term rainfall data by 1D CNN, combined with other driving data by LSTM, would improve the simulation accuracy of the model.

Experiments 2 and 4 are 'n time step output' simulations, which means that these two experiments are used to simulate future multi-day discharges. In Experiment 2 we use two different types of data for simulation. We consider 5 different look-back windows, 7 days, 15 days, 30 days, 180 days, and 365 days and two-look forward windows, which are 3 days and 5 days.

Experiment 4 is to test whether the proposed LSTM+1D CNN model can improve the simulation accuracy of the 'n time step output' simulation. We consider 2 different look-back windows for rainfall, which are 10 days and 3 days, and 3 different look-back windows for other input, which are 30 days, 180 days and 365 days.

## 3 Results and Discussion

### 3.1 Comparison of the results from different types of rainfall driven data for 'one time step output' simulation
**(Experiment 1)**

We first compared the model results for different look-back windows driven by different types of rainfall data. Figure 4 and Figure 5 show the simulated discharge process for the two catchments. The simulation results of the hydrological model are also placed in each figure for comparison. As can be seen from the figures, for both catchments, the simulation processes of the deep learning model are closer to the measured processes when compared to the hydrological model. In particular, for the

second catchment, the simulation results of the deep learning model are significantly better than the simulation results of the hydrological model.

Table 2 shows the performance of Experiment 1 using different types of rainfall and different look-back windows. As can be seen from the table, for Catchment 1, the smallest error is obtained for 7 days as the look-back window when driven by basin





mean rainfall data, where the RMSE is 0.309542 and the NSE is 0.947387. By increasing the length of the look-back window,
the change of error does not show a gradual decrease trend. The maximum error of the model is obtained when the length of
the look-back window is 180 days, where the RMSE is 0.351222 and the corresponding NSE is 0.932265. For Catchment 2,
the simulation results of the model tend to get better gradually when the length of the look-back window is increased. The best
result is obtained when the look-back window is 365 days, where the RMSE is 0.200881 and the NSE is 0.992691. The worst
results are obtained when the look-back window is 7 days, and the results for the look-back window of 30 days are slightly
worse than those for the look-back window of 15 days.

If we consider the simulation results driven by spatially distributed rainfall data, for Catchment 1, when the look-back windows
are 7, 15, 30, and 180, spatially distributed rainfall data are better than those driven by the basin mean rainfall data model.
Among them, the model simulates best when the look-back window is 30 days, where the RMSE is 0.278799 and the NSE is
0.957319. For Catchment 2, the simulation results driven by spatially distributed rainfall data are better than those driven by
the basin mean rainfall data model when the look-back windows are 7, 15, and 30. Among them, the model simulates best
when the look-back window is 7 days, where the RMSE is 0.204561 and the NSE is 0.992421.

With the simulation results of the two different types of driving data, we cannot conclude about which look-back window can
achieve the best simulation results for both catchments. This means that when we use LSTM for rainfall-runoff simulation, we
need to compare different look-back windows to obtain the best simulation results. However, the results driven by spatially
distributed rainfall data from both basins show that when the look-back windows are small (7 days,15 days, 30 days), the
results obtained by spatially distributed rainfall data are better than those when the look-back windows are large.

We also compared all the results from the deep learning model with the results from the traditional hydrological model. For
Catchment 1, the hydrological model simulation result has an RMSE of 0.618972 and an NSE of 0.789625. For Catchment 2,
the hydrological model simulation result has an RMSE of 0.97845 and an NSE of 0.826605. For both catchments, the deep
learning models can achieve better results regardless of which look-back window and drive data are used. The NSE of all of
them exceeds 0.9, which means that LSTM can simulate the whole discharge process better than hydrological model.

Table 2. Comparison of performance of Experiment 1 using different types of rainfall and different look-back windows

| Look-back windows | Catchment 1 | | | | | |
| | Catchment 1 Driven by the basin mean rainfall data | | | Driven by spatially distributed rainfall data | | |
| | NSE | RMSE (mm/d) | EPD | NSE | RMSE (mm/d) | EPD |
| 7 days | 0.947387 | 0.309542 | 11.37 | 0.956200 | 0.282431 | -0.206 |
| 15 days | 0.932467 | 0.350697 | 7.791 | 0.951584 | 0.296939 | 1.88 |
| 30 days | 0.946275 | 0.312798 | 9.198 | 0.957319 | 0.278799 | 4.703 |
| 180 days | 0.932265 | 0.351222 | 3.582 | 0.940281 | 0.329786 | 7.084 |
| 365 days | 0.938822 | 0.333790 | 11.46 | 0.932898 | 0.349576 | 8.36 |
| Look-back windows | Catchment 2 | | | | | |
| | Driven by the basin mean rainfall data | | | Driven by spatially distributed rainfall data | | |
| | NSE | RMSE (mm/d) | EPD | NSE | RMSE (mm/d) | EPD |
| 7 days | 0.987646 | 0.261172 | 4.674 | 0.992421 | 0.204561 | 7.394 |



| | | | | | | |
|---|---|---|---|---|---|---|
| 15 days | 0.989722 | 0.238217 | 3.521 | 0.992108 | 0.20875 | 5.837 |
| 30 days | 0.989680 | 0.238708 | 11.02 | 0.991442 | 0.217378 | 2.383 |
| 180 days | 0.991418 | 0.217678 | 10.41 | 0.991226 | 0.220095 | 6.13 |
| 365 days | 0.992691 | 0.200881 | 10.27 | 0.988555 | 0.251379 | 6.337 |

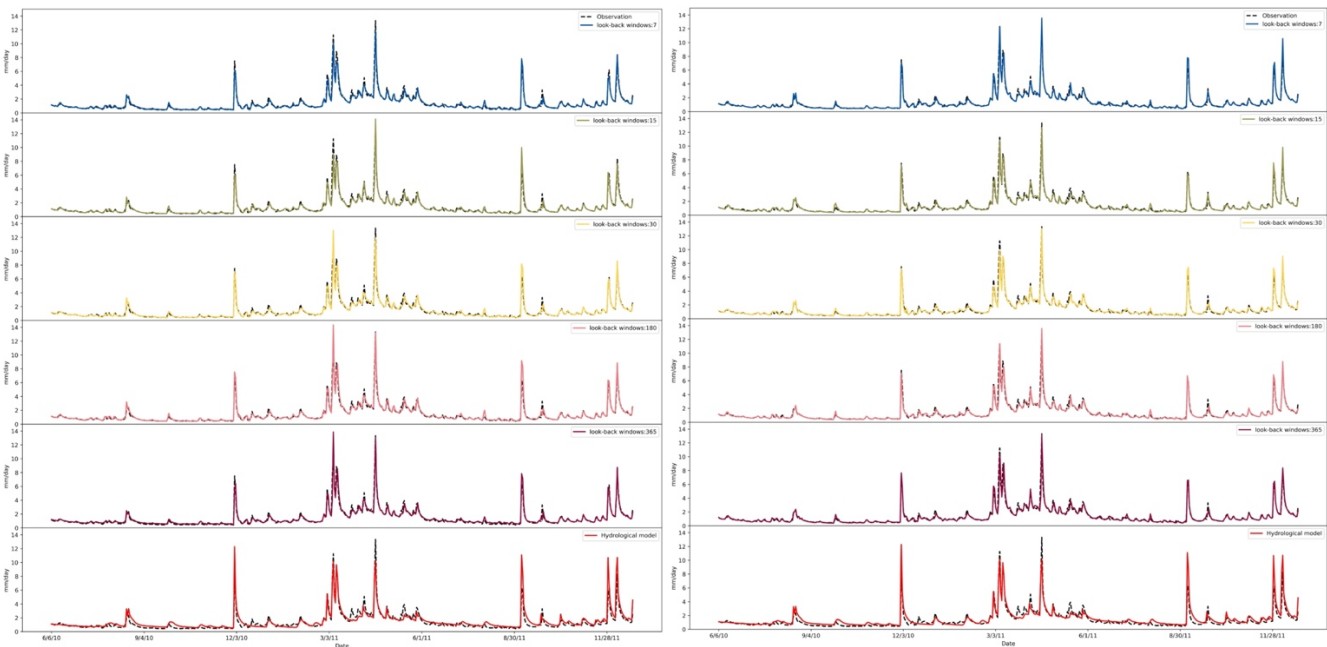

Figure 4. the results of Experiment 1 using different type of rainfall data for Catchment 1(left: Driven by the basin mean rainfall data; right: Driven by spatially distributed rainfall data





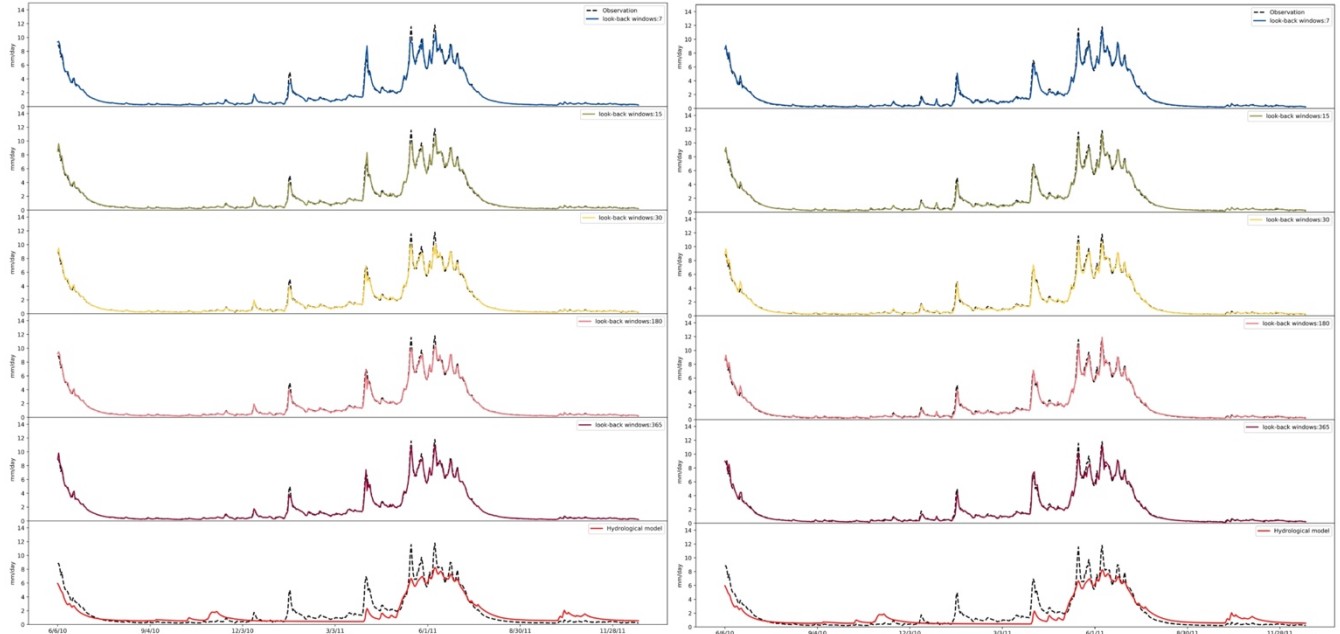

Figure 5. the results of experiment 1 using different type of rainfall data for Catchment 2(left: Driven by the basin mean rainfall data; right: Driven by spatially distributed rainfall data

## 3.2 Comparison of the results from different types of rainfall driven data for 'n time step output' simulation (Experiment 2)

In Experiment 2 we tested the simulation ability of LSTM for n time steps output. Figure 6 and Figure 7 show the simulation ability of 3-time steps output and Figure 8 and Figure 9 show the simulation ability of 5- time steps output. By comparing the model results with those of Experiment 1, which is the 1-time step output, we find that the results become worse whether driven by the basin mean rainfall data or by spatially distributed rainfall data. For 3-day look-forward windows simulation of Catchment 1, the model driven by spatially distributed rainfall data outperforms the model driven by basin mean rainfall data for each look-back window. The results (Table 3) show that the model driven by the basin mean rainfall data with 30 days look-back window has the highest NSE of 0.927501 and the lowest RMSE of 0.363362. The model driven by the spatially distributed rainfall data with 7 days look-back window has the highest NSE of 0.943605 and the lowest RMSE of 0.320477. For Catchment 2, the RMSE of the model driven by the spatially distributed rainfall data is 0.336385 and 0.321342 when the look-back windows are 7 and 15 days, respectively, both of which are better than the results obtained by the basin mean rainfall data. However, when the look-back window continues to increase, the results driven by the basin mean rainfall data are better than those driven by the spatially distributed rainfall data. For example, when the look-back window is 365, the RMSE obtained with the input of basic mean rainfall data is 0.309905 and the corresponding NSE is 0.982605. The RMSE obtained with the input of spatially distributed rainfall data is 0.384379 and the NSE is 0.973240. However, when we focus on the results with the look-forward window of 5 days, we find that the results driven by spatially distributed rainfall data at every look-back





window are better than the results driven by the mean. For Catchment 1, the best result is obtained at the look-back window of 7 days. This result is obtained at a look-back window of 30 days for Catchment 2. Although the simulation results with 'n

time step output' are less effective than that with '1 time step output', if we compare the results with those of the hydrological model, the simulation with LSTM for 'n time step output' still gives reliable results. The NSE of all the results is greater than 0.9, indicating that LSTM can still simulate the discharge process very well even with the 'n time step output' simulation, regardless of the type of driver data used.

We compared the results of basin mean rainfall data driven and spatially distributed rainfall data driven simulations at 1 time

step output, 3-time steps output, and 5-time steps output. We find that, in general, the results of the LSTM model are slightly better than those driven by the surface mean rainfall data by adding the spatial distribution information of rainfall. In particular, the model results driven by spatially distributed rainfall data are the best in both catchments when the look-back windows are 7 days and 15 days. Increasing the length of the look-back window leads to slightly worse results than those driven by basin mean rainfall data, but the difference between the two is not significant.

310       Table 3. Comparison of performance of Exp. 2 using different types of rainfall, different look-back, and look-forward windows

| Look-back windows | Look-forward windows | Catchment 1 | | | | | |
| --- | --- | --- | --- | --- | --- | --- | --- |
| | | Driven by the basin mean rainfall data | | | Driven by spatially distributed rainfall data | | |
| | | NSE | RMSE (mm/d) | EPD | NSE | RMSE (mm/d) | EPD |
| 7 days | 3 days | 0.924755 | 0.370180 | 16.98 | 0.943605 | 0.320477 | 1.659 |
| 15 days | | 0.91966 | 0.382509 | 12.17 | 0.929370 | 0.358648 | 1.432 |
| 30 days | | 0.927501 | 0.363362 | 3.475 | 0.939554 | 0.331787 | 7.573 |
| 180 days | | 0.923172 | 0.374054 | 11.62 | 0.938786 | 0.333887 | 7.043 |
| 365 days | | 0.923299 | 0.373744 | 7.326 | 0.934829 | 0.344509 | 10.57 |
| 7 days | 5 days | 0.928207 | 0.361588 | 14.5 | 0.937041 | 0.338613 | 3.636 |
| 30 days | | 0.915322 | 0.392699 | 2.558 | 0.927210 | 0.364090 | 4.548 |
| 365 days | | 0.897266 | 0.432545 | 4.936 | 0.925794 | 0.367615 | 4.243 |
| Look-back windows | Look-forward windows | Catchment 2 | | | | | |
| | | Driven by the basin mean rainfall data | | | Driven by spatially distributed rainfall data | | |
| | | NSE | RMSE (mm/d) | EPD | NSE | RMSE (mm/d) | EPD |
| 7 days | 3 days | 0.978644 | 0.343388 | 19.61 | 0.979506 | 0.336385 | 8.251 |
| 15 days | | 0.980120 | 0.331303 | 12.3 | 0.985025 | 0.321342 | 11.41 |
| 30 days | | 0.981325 | 0.321110 | 13.47 | 0.978077 | 0.347910 | 13 |
| 180 days | | 0.980785 | 0.325719 | 10.94 | 0.978526 | 0.344328 | 9.777 |
| 356 days | | 0.982605 | 0.309905 | 8.114 | 0.973240 | 0.384379 | 9.645 |
| 7 days | 5 days | 0.973028 | 0.385905 | 17.65 | 0.976380 | 0.361128 | 7.743 |
| 30 days | | 0.977469 | 0.352251 | 14.56 | 0.982341 | 0.312704 | 9.697 |
| 365 days | | 0.978389 | 0.345585 | 10.58 | 0.979603 | 0.325428 | 10.24 |





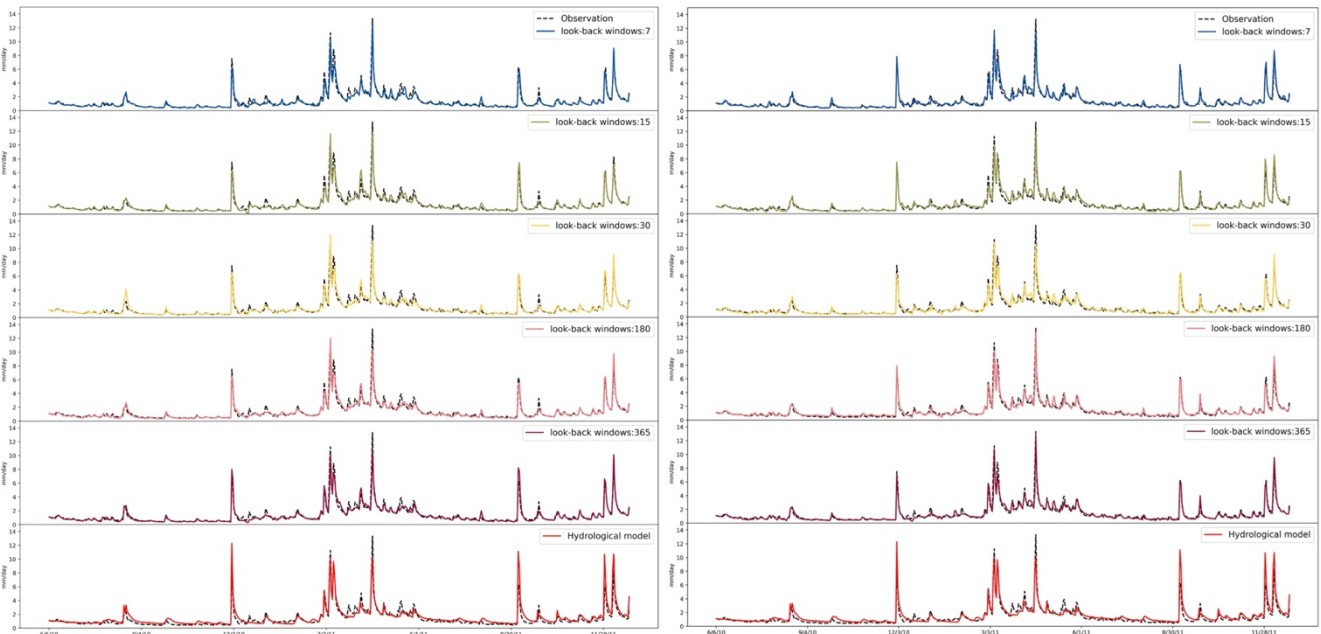

Figure 6. the results of Experiment 2 (3 days look-forward windows) using different type of rainfall data for Catchment 1(left: Driven by the basin mean rainfall data; right: Driven by spatially distributed rainfall data)


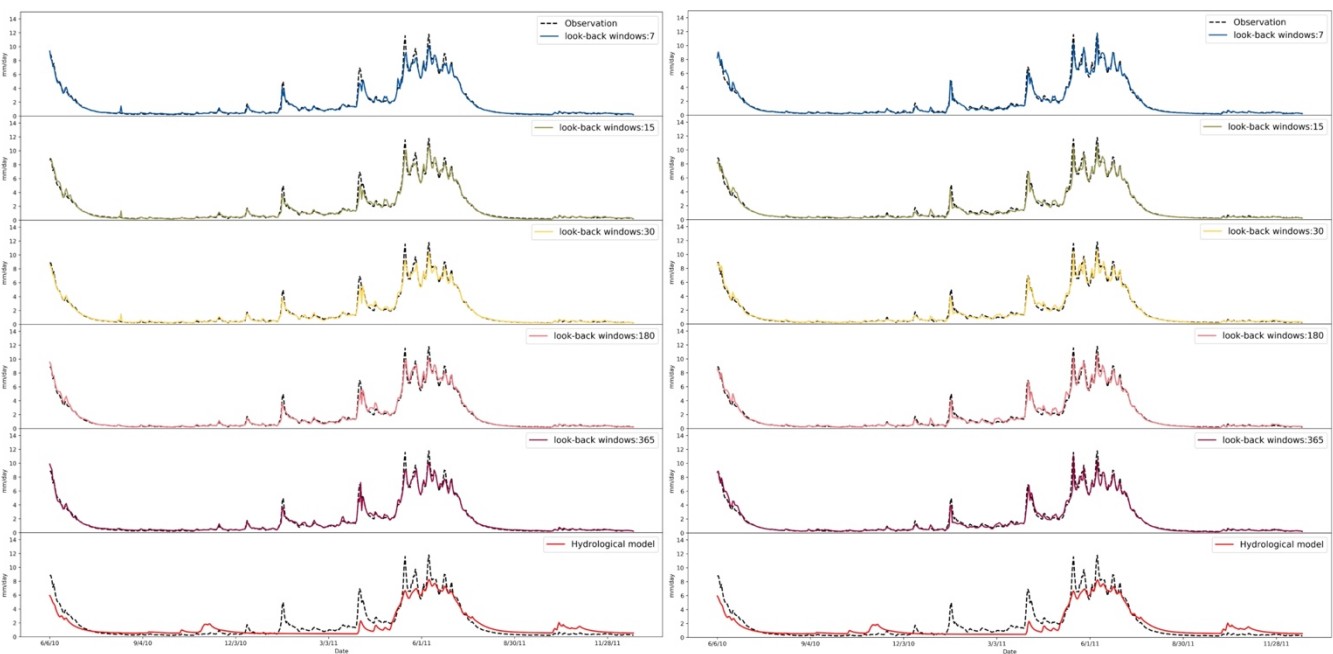

Figure 7. the results of Experiment 2 (3 days look-forward windows) using different type of rainfall data for Catchment 2(left: Driven by the basin mean rainfall data; right: Driven by spatially distributed rainfall data)






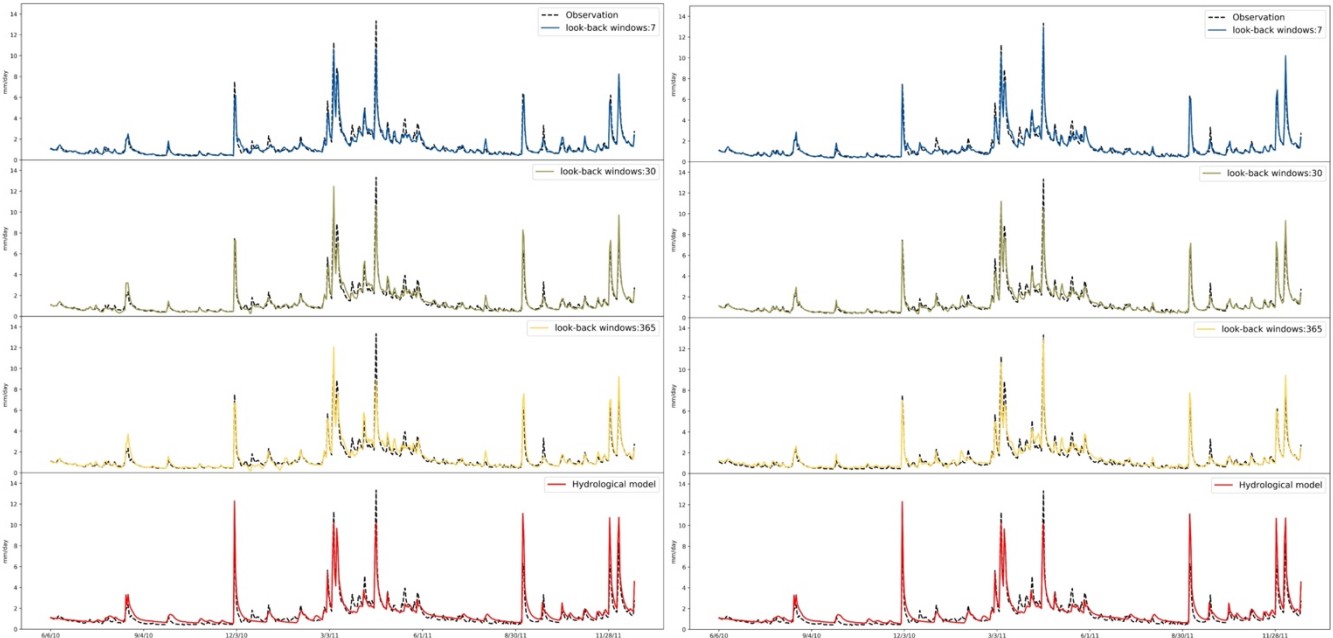

Figure 8. the results of Experiment 2 (5 days look-forward windows) using different type of rainfall data for Catchment 1(left: Driven by the basin mean rainfall data; right: Driven by spatially distributed rainfall data)

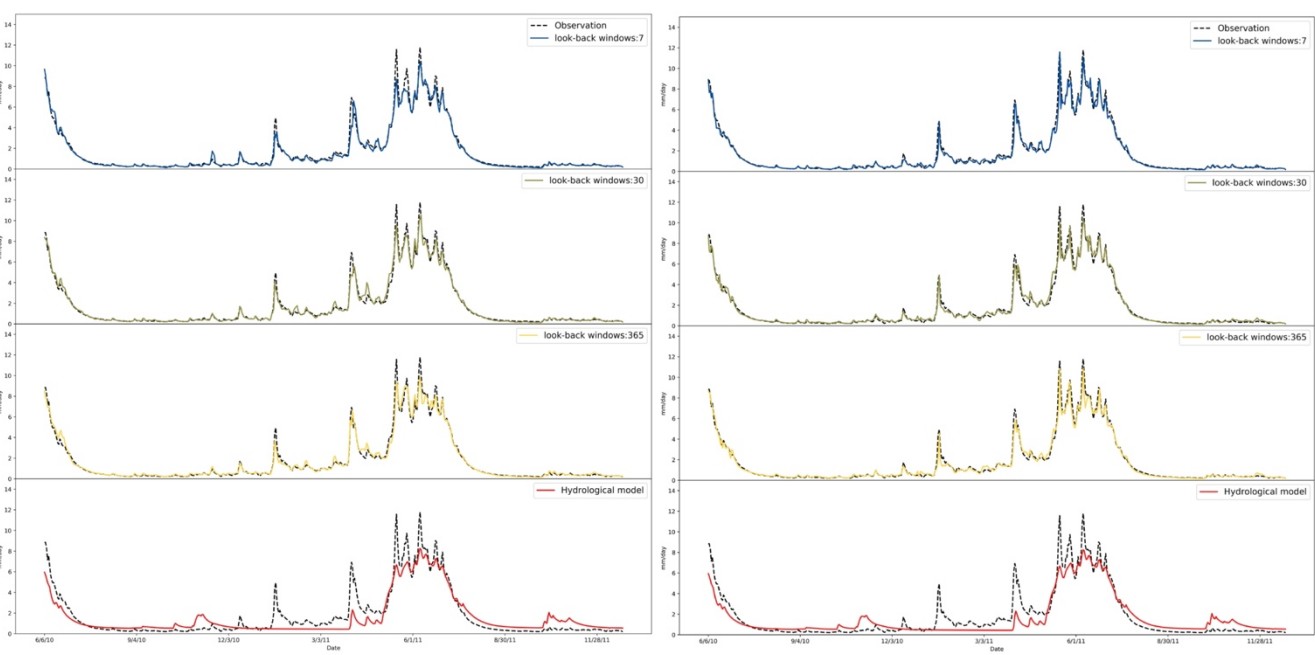


Figure 9. the results of Experiment 2 (5 days look-forward windows) using different type of rainfall data for Catchment 2(left: Driven by the basin mean rainfall data; right: Driven by spatially distributed rainfall data)





### 3.3 Results of simulation with LSTM+1D CNN for 'one time step output' simulation (Experiment 3)

By observing the simulation results of traditional LSTM on spatially distributed rainfall data and basin mean rainfall data, we find that using the spatial distribution information of rainfall in shorter look-forward windows can play a better role. Therefore, in Experiment 3, we simulated 'one time step output' by the proposed LSTM+1D CNN. Our look-forward windows for rainfall information are chosen to be 3 days and 5 days, and the look-forward windows for other input data are chosen to be 30 days, 180 days, and 365 days. Performance of both catchments are shown in Table 4.

Table 4. Comparison of performance of Exp. 3 using shorter look-back window for rainfall and long look-back window for other data

| Look-back windows for spatial rainfall | Look-back windows for other inputs | Catchment 1 | | |
|---|---|---|---|---|
| | | NSE | RMSE (mm/d) | EPD |
| 3 | 30 | 0.938193 | 0.335501 | 6.816 |
| 3 | 180 | 0.919809 | 0.382152 | 8.646 |
| 3 | 365 | 0.934796 | 0.344598 | 6.135 |
| 10 | 30 | 0.926808 | 0.365095 | 16.74 |
| 10 | 180 | 0.922589 | 0.375472 | 11.08 |
| 10 | 365 | 0.917416 | 0.387814 | 16.1919 |
| Look-back windows for spatial rainfall | Look-back windows for other inputs | Catchment 2 | | |
| | | NSE | RMSE (mm/d) | EPD |
| 3 | 30 | 0.989939 | 0.235687 | -1.109 |
| 3 | 180 | 0.992046 | 0.209567 | 3.433 |
| 3 | 365 | 0.992745 | 0.20014 | 7.733 |
| 10 | 30 | 0.991547 | 0.216041 | 5.16 |
| 10 | 180 | 0.990718 | 0.22638 | 5.075 |
| 10 | 365 | 0.990619 | 0.227585 | 3.59727 |

For Catchment 1, the RMSEs obtained by the proposed LSTM+1D CNN are 0.335501 and 0.365095 when the look-back window for other inputs is 30 days, corresponding to 3 and 10 days of look-back windows. We know from previous

experiments that when the look-back window is 30 days, the result driven by spatially distributed rainfall data is 0.278799 and the result driven by basin mean rainfall data is 0.312798. LSTM+1DCNN is close to the result obtained by basin mean rainfall data. Similarly, we find that for other combinations of look-back windows, the simulation results of the proposed LSTM+1D CNN are worse than those in Experiment 1.

For Catchment 2, the RMSEs obtained by the proposed LSTM+1D CNN are 0.235687and 0.216041when the look-back

window for other inputs is 30 days, corresponding to 3 and 10 days of look-back windows. Both results are worse than those driven by spatially distributed rainfall data in Experiment 1, but slightly better than those driven by the basin mean rainfall data. The results obtained by the proposed LSTM+1D CNN when the look-back window for other inputs is 180 days are also between the results obtained for the different driving data in Experiment 1. The same results were also found when the look-back window is 365 days.






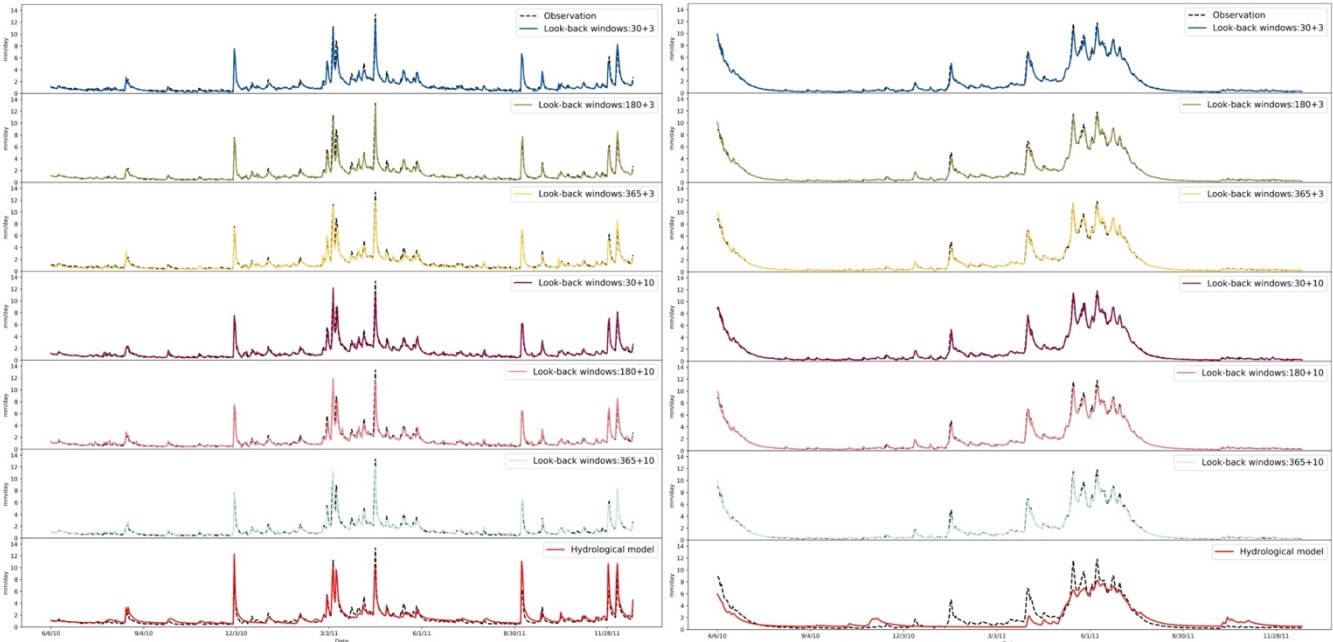

Figure 10. the results of Experiment 3 using LSTM+1DCNN for two catchments (left: Catchment1; right: Catchment 2)

## 3.4 Results of simulation with LSTM+1D CNN for 'n time step output' simulation (Experiment 4)

We also performed the 'n time step output' simulation using the proposed LSTM+1D CNN. In order to compare the results in Experiment 2, the look-forward windows chosen were 3 and 5 days. Look-back windows for spatial rainfall were 3 and 10

days, and look-back windows for other inputs were set to 30, 180 and 365 days.

For Catchment 1, the results are worse than those of the traditional LSTM model for both the future 3 days and the future 5 days simulations by the proposed LSTM+1D CNN. When the results are driven by the basin mean rainfall data, they are not much different from those obtained by the conventional LSTM. For Catchment 2, the proposed LSTM+1D CNN gives better results for the next 3 days than LSTM's. For the 5-day future simulation, the proposed LSTM+1D CNN performs better than

when the results are driven by the basin mean rainfall data, but when the look-back windows are 30 and 365 days, the results of the proposed LSTM+1DCNN are slightly worse than the spatial ones. The results are slightly worse than those driven by spatially distributed rainfall data. Comparing the results of different rainfall look-back windows for the same the look-back windows for other inputs, we find that the results for the look-back window of 10 are generally slightly better than those for the look-back window of 3, except for the simulation of Catchment 2 for the next 5 days.

Although the simulation results of the proposed model are not completely better than LSTM driven by spatially distributed rainfall data, the results of the model are comparable to those driven by basin mean rainfall data. This is because the vector representing the spatial distribution information of rainfall in Catchment 2 is longer than that in Catchment 1, and the results of the proposed LSTM+1D CNN in Catchment 2 are better than those in Catchment 1. We speculate that increasing the spatial





distribution information of rainfall by increasing the resolution can improve the simulation results of the proposed LSTM+1D

CNN.

Table 5. Comparison of performance of Exp. 3 using shorter look-back window for rainfall and long look-back window for other data

| Look-back windows for spatial rainfall | Look-back windows for other inputs | Look-forward windows | Catchment 1 | | |
|---|---|---|---|---|---|
| | | | NSE | RMSE (mm/d) | EPD |
| 3 | 30 | 3 days | 0.902964 | 0.410379 | 5.009 |
| 3 | 180 | | 0.908226 | 0.399513 | 9.188 |
| 3 | 365 | | 0.910881 | 0.393547 | 5.838 |
| 10 | 30 | | 0.902725 | 0.410897 | 1.474 |
| 10 | 180 | | 0.910135 | 0.394548 | 3.252 |
| 10 | 365 | | 0.91415 | 0.385408 | 4.997 |
| 3 | 30 | 5 days | 0.912814 | 0.398473 | 3.693 |
| 3 | 180 | | 0.911967 | 0.400404 | 0.3612 |
| 3 | 365 | | 0.914056 | 0.395623 | 6.18 |
| 10 | 30 | | 0.919725 | 0.382354 | 0.2683 |
| 10 | 180 | | 0.926335 | 0.366274 | 5.731 |
| 10 | 365 | | 0.912441 | 0.399325 | 4.116 |
| Look-back windows for spatial rainfall | Look-back windows for other inputs | Look-forward windows | Catchment 2 | | |
| | | | NSE | RMSE (mm/d) | EPD |
| 3 | 30 | 3 days | 0.988604 | 0.250838 | 5.16 |
| 3 | 180 | | 0.987514 | 0.262566 | 5.075 |
| 3 | 365 | | 0.98514 | 0.286436 | 3.59727 |
| 10 | 30 | | 0.989325 | 0.242774 | -1.109 |
| 10 | 180 | | 0.990405 | 0.23017 | 3.433 |
| 10 | 365 | | 0.98718 | 0.266047 | 7.733 |
| 3 | 30 | 5 days | 0.979094 | 0.33975 | 10.65 |
| 3 | 180 | | 0.978882 | 0.341467 | 11.2 |
| 3 | 365 | | 0.984175 | 0.295589 | 13.92 |
| 10 | 30 | | 0.981248 | 0.321771 | 14.59 |
| 10 | 180 | | 0.976705 | 0.358638 | 5.89 |
| 10 | 365 | | 0.980956 | 0.324266 | 17.15 |



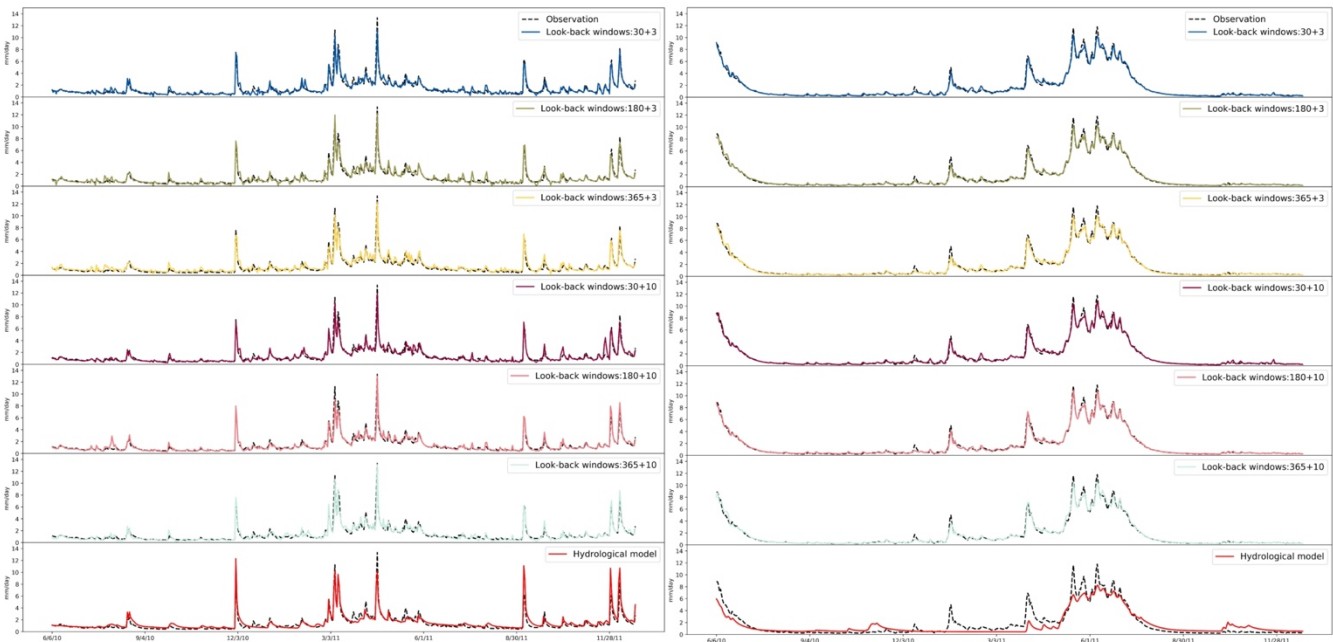

Figure 11. the results of Experiment 4 using LSTM+1DCNN for 3-time steps output (left: Catchment1; right: Catchment 2)


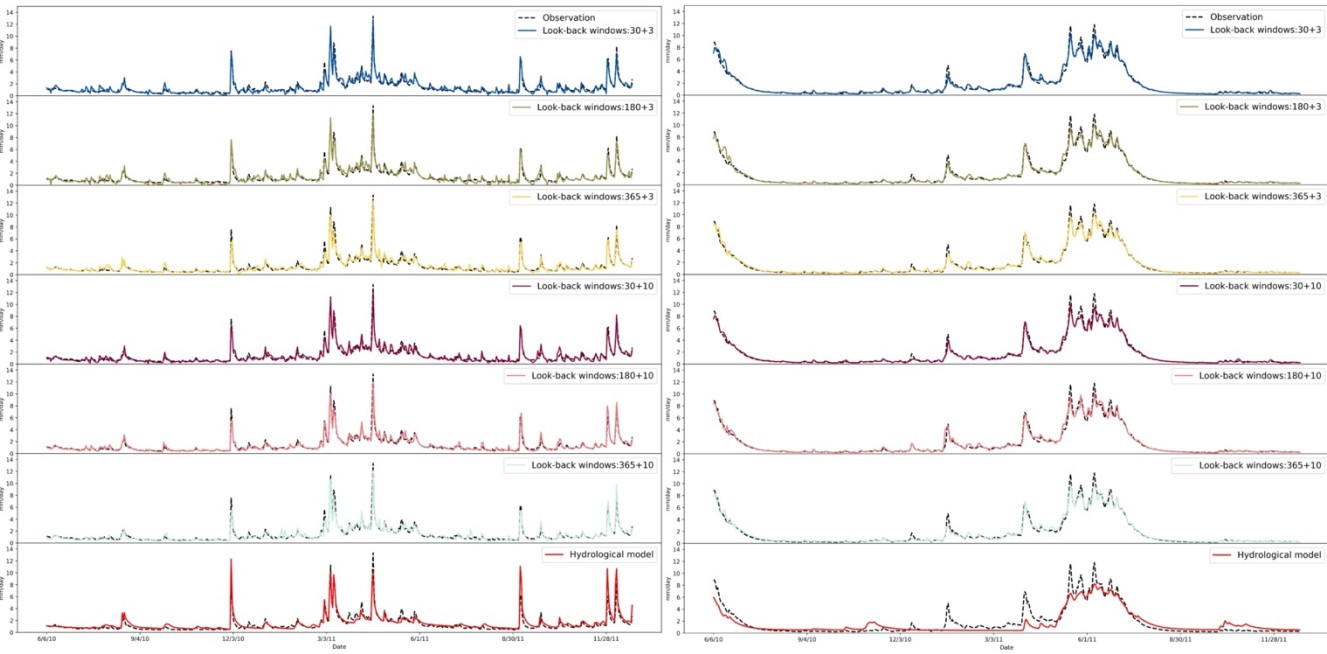

Figure 12. the results of Experiment 4 using LSTM+1DCNN for 5-time steps output (left: Catchment1; right: Catchment 2)





## 4 Conclusions and Future Research

Deep learning models, especially LSTM, have received increasing attention in rainfall-runoff simulation studies. The current LSTM-based studies are still mainly from a data-driven perspective and few studies have investigated the different simulation results from different types of meteorological data or construction of models based on the physical relationships of rainfall and runoff.

In this study, rainfall, which has the greatest influence on runoff, is used as the object of study. The basin mean rainfall data is
used as the rainfall data without spatial distribution information, and the vector composed of rainfall on hydrologic response units in the basin is used as the rainfall data with spatial distribution information. The impact of the two types of rainfall data on the performance of the deep learning model is compared and analyzed.

According to the results of Experiment 1, when using LSTM for the simulation of 'one time step output' adding the spatial distribution information of rainfall, which means that driven by spatially distributed rainfall data, can slightly improve the
performance of the LSTM model. When the look-back windows are 7 and 15 days, the results obtained using spatially distributed rainfall data are significantly better than those obtained using basin mean rainfall data. For the simulation of peak discharge, adding the spatially distributed rainfall data can significantly improve the simulation results of the LSTM model.

The same conclusion can also be obtained from Experiment 2, which is for 'n time step output' simulation, adding the spatial distribution information of rainfall can improve the LSTM model for look-forward windows of 3 and 5 days. As in Experiment
1, the results driven by spatially distributed rainfall data are significantly better than those driven by basin mean rainfall data when the look-back windows are 7 and 15 days. For the simulation of 'n time step output', adding the spatially distributed rainfall information can still improve the model's simulation of peak discharge.

Considering that the spatial distribution information of increased rainfall performs better in shorter look-back windows, the study proposes the LSTM+1D CNN model. In the model, the meteorological data and discharge of longer time series are
processed by the LSTM model, and the rainfall data of shorter time series are processed by 1D CNN. The output of the final model is a combination of the results of the LSTM model, the results of the 1D CNN model, and the rainfall of the day. The simulation results of the model for 'n time step output' and 'one time step output' are examined in Experiment 3 and Experiment 4, respectively. Although the simulation results of the proposed model are not completely better than the LSTM driven by spatially distributed rainfall data, the results of the model are comparable to those driven by basin mean rainfall data. Since
the simulation effect of the proposed LSTM+1D CNN in Catchment 2 is better than that of Catchment 1, and the spatial distribution information of rainfall in Catchment 2 is more abundant, we guess that increasing the spatial distribution information of rainfall can improve the simulation results of the proposed LSTM+1D CNN model.

For different experiments, the results of the deep learning model are better than the physical model. Since the study only compares the simulation results of the lumped hydrological model in the data, we cannot conclude that the simulation results
of the deep learning model are better than the physical model in both catchments. However, the experimental results demonstrate the great potential of deep learning models for rainfall runoff simulation.



In summary, we did not find a certain look-back window which is optimal for different watersheds and different types of simulations. This means that different look-back windows should be explored to obtain the optimal results when using LSTM models for relevant simulations. Adding the spatial distribution information of rainfall can improve the simulation results of
the LSTM model, and this improvement is more obvious under the condition of the short look-back window. Adding the spatial distribution information of rainfall can dramatically improve the simulation results of the LSTM model for peak discharge. The results of our proposed LSTM+1D CNN on 'n time step output' and 'one time step output' are comparable to those of the LSTM model driven by basin mean rainfall data, and slightly worse than those of spatially distributed rainfall data.

Although our proposed LSTM+1D CNN does not significantly perform better than the LSTM model, it is still useful for
subsequent related studies, especially flood prediction. Since raster rainfall data with spatial distribution information are currently available from many sources, we can use these data to drive deep learning models without looking for longer series rainfall data, which is in some catchments, to obtain comparable runoff simulation results.

There are some gaps that can be continued to be investigated in the future. For example, in this study, the rainfall of the hydrological response unit of catchment is used to represent the spatial distribution of rainfall information. We can obtain
raster-type rainfall data from satellite data, climate models, and other sources, which may be able to better represent the spatial distribution of rainfall. In addition, the use of raster type rainfall data can help us use 2D CNN instead of 1D CNN, which can better characterize the spatial distribution of rainfall. In this study, we only consider comparing the basin mean rainfall and spatially distributed rainfall, other driving data, such as temperature and a pressure, also have spatial distribution characteristics. How to increase the spatial distribution information of all features on the basis of the uniform resolution of
different features and compare the influence of the input conditions on the model results is also a research direction worth conducting in the future.






## Appendix A: Hyperparameter tuning

The values of the main parameters for different experiments are shown in the following table.


Table Setting of main parameters for different experiments

| ID | Type of rainfall | Catchment 1 | Catchment 2 |
|---|---|---|---|
| Exp. 1 | basin mean rainfall data | LSTM: Input size=6, hidden size=256, num layers=1, dropout rate=0.3, batch size = 64, epochs = 200 | LSTM: Input size=6, hidden size=256, num layers=1, dropout rate=0.3, batch size = 64, epochs = 200 |
| | spatially distributed rainfall | LSTM: Input size=69, hidden size=256, num layers=1, dropout rate=0.3 batch size = 64, epochs = 200 | LSTM: Input size=199, hidden size=256, num layers=1, dropout rate=0.3 batch size = 64, epochs = 200 |
| Exp. 2 | basin mean rainfall data | LSTM: Input size=6, hidden size=256, num layers=1, dropout rate=0.3 batch size = 64, epochs = 200 | LSTM: Input size=6, hidden size=256, num layers=1, dropout rate=0.3, batch size = 64, epochs = 200 |
| | spatially distributed rainfall | LSTM: Input size=69, hidden size=256, num layers=1, dropout rate=0.3 batch size = 64, epochs = 200 | LSTM: Input size=199, hidden size=256, num layers=1, dropout rate=0.3 batch size = 64, epochs = 200 |
| Exp. 3 | spatially distributed rainfall | LSTM: Input size=5, hidden size=256, num layers=1, dropout rate=0.3 batch size = 64, epochs = 200 CNN: conv1:(look-back windows+1, 16,3, padding=1), conv2:(16, 32,3, padding=1), max pool=3 | Batch size = 64, Epochs = 200 LSTM: Input size=5, hidden size=256, num layers=1, dropout rate=0.3 CNN: conv1:(look-back windows+1, 16,3, padding=1), conv2:(16, 32, 3, padding=1), max pool=3 |
| Exp. 4 | spatially distributed rainfall | LSTM: Input size=5, hidden size=256, num layers=1, dropout rate=0.3 batch size = 64, epochs = 200 CNN: conv1:(look-back windows+1, 16, 3, padding=1), conv2:(16, 32, 3, padding=1), max pool=3 | Batch size = 64, Epochs = 200 LSTM: Input size=5, hidden size=256, num layers=1, dropout rate=0.3 CNN: conv1:(look-back windows+1, 16,3, padding=1), conv2:(16, 32, 3, padding=1), max pool=3 |




*Code and data availability.* The CAMELS input data are freely available at the homepage of the NCAR (https://ral.ucar.edu/solutions/products/camels). Model outputs as well as code may be made available by request to the corresponding author.

*Author contributions*. YW and HK jointly developed the project idea and performed research. YW proposed the LSTM+1D
CNN architecture and conducted all the experiments and analyzed the results. HK supervised the manuscript from the machine-learning perspective. All authors read and approved the final manuscript.

*Competing interests*. The authors declare that they have no conflict of interest.

*Acknowledgements*. Not applicable.

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
