# Peer review of "Impact of Spatial Distribution Information of Rainfall in Runoff Simulation Using Deep-Learning Method"

_Hydrology and Earth System Sciences, 2021_

## Author Response (AR1)

**Authors' response for Anonymous Referee #1**

1. setting:

Thank you very much for your suggestion. Your suggestion about the setting is very helpful for me to revise my paper. Most studies apply rainfall and previous discharges with different time steps and combinations as inputs. But as you said, if we put the discharge in the input, the relationship between it and the output would affect the analysis of the effect of different types of rainfall on the results. So in the new version, I removed the discharge from the inputs. I think what you mentioned about using LSTM as an individual model and regional model is a very interesting aspect. In the new version, I added the number of catchments to 10. I compared the difference between LSTM as an individual model trained separately for each catchment and as a regional model trained at the same time for multiple catchments.

2 Method:

Thanks for the suggestion. I have gone into more detail in the data section.

As can be seen in Figure 1, instead of using the catchment mean rainfall data (see the top of Figure 1b), we extract the rainfall of all hydrologic response units in the catchment to form a vector. The bottom of Figure1b shows that the catchment has 8 hydrologic response units from which we extract the corresponding 8 rainfall data to form a vector of size 8. Since the values in this vector represent rainfall information at different locations in the catchment, our assumption is that the vector is rainfall data with spatial distribution information.

[Figure]

Figure1. a: Ten catchments and their locations in the State; b: Examples of spatially distributed rainfall data in this study

3 Results:

Thanks for the suggestion. I modified the number of digits in the result to two digits. For Experiment 1, I compared the results of different types of rainfall data when LSTM was used as individual model for one-time step output; For Experiment 2, I compared the results of different types of rainfall data when LSTM was used as regional model for one-time step output; In three experiments, we compare the effect of different types of rainfall types on individual model as well as regional model when simulating n time steps output.

**Authors' response for Anonymous Referee #2**

1. Thanks for the suggestion. I modified the number of digits in the result to two digits.

2. I have added basic information about the catchment in Methods and Dataset.

3-5. Thank you for the suggestion. I changed the data splitting to a 70-2-10 split for each catchment.

Thank you for the suggestion. For better analysis and comparison, I increase the number of catchments to 10. In data and method, I provide the statistics of each catchment. In the new version of the conclusions, I compare the results in various ways.

6. In our conclusion, we also obtained similar results to other studies that a longer look back window leads to better results, e.g., 365 days. I think it is very interesting to analyze the relationship between the basic properties of the watershed and the look-back window based on these conclusions.

7. 180 and 365 days as look-back windows are often used in other studies that apply deep learning models to the field of hydrology—considering the advantages of LSTM models, as data-driven models, which discover the changing patterns of time series data. We can assume that 180 and 365 days as look-back windows help the model learn the correlation between long series of rainfall, runoff, and other factors. Data-driven models can handle longer windows, which can provide more information. How to choose look-back windows is a question that needs to be further investigated. This is the reason why we compare different windows in the paper.

8. I have added basic information about the catchment in Methods and Dataset.

9. Thank you for the suggestion. In the new version, with the addition of more catchments, I have added new comparison plots instead of just flow processes. For the flow process plots, I kept the comparisons between models with the most significant differences in results.

10. Thank you for the reminder. All the variables and functions are explained in the corresponding places.

11. It should be "rain gauge" or "rain station." Thank you for the suggestion.

12. It should be "activation function." Thank you for the suggestion.

---

## Referee Report (RR1)

Review
Impact of Spatial Distribution Information of Rainfall in Runoff Simulation Using Deep-Learning Methods
Author(s): Yang Wang and Hassan A. Karimi
Submitted to HESS

2nd review

Review date: 24 Feb 2022

The authors have done a good job in updating the manuscript. I still have one suggestion and some comments

1. Please zoom in one (or two) pick flow and present the variation of simulated and observed flows along with the rainfall used in the simulation. Please choose a long look back window. It will be useful to see if there are any lags or not (which cannot be seen clearly in Fig 7). It is important to show that the model(s) built is sensitive to rainfall and does not suffer from the influence of long look back window(s).

2. OTHER COMMENTS
Please read and edit carefully the manuscript as some of the figures and tables are not numbered correctly. Some suggestions are:
   a. Abstract: "regional" instead of region models
   b. Line 168: Unclear and please rephrase it: "Since the values in this vector represent rainfall information at different locations in the catchment, our assumption is that the vector is rainfall data with spatial distribution information."
   c. Line 206: Clarify Snow-17 models
   d. Line 289:  "… 70% of the data are used for model training, 20% for model validation, and 10% for model testing." I think validation and testing are used synonymously. Perhaps you want to use the term cross-validation instead of validation. If it is so then the correct sentence should be: "… 70% of the data are used for model training, 20% for model testing, and 10% for model cross-validation."
   e. Line 294: " ... catchment 1-5 are combined to train regional 295 model 1, and training data from catchment 6-10 are combined to train regional model 2." Fig 1 does not show catchment numbers. Or have I missed where you have shown the numbers of the catchments?
   f. Line 304:"is" to was
   g. Line 303: Table 1 should be Table 2
   h. Table 2: Define D

---

## Author Response (AR2)

**Authors' response for Anonymous Referee #1**

1. First of all, I would like to commend the authors for the hard work and – as a result – great improvements they could achieve on the manuscript. When reading the provided answers to my first review I was not sure if the authors did understand my critique. However, their revisions prove me wrong. What I am still missing is a discussion about why a "forecasting" setting was chosen over a "simulation" setting for the examination — albeit the former somehow undermines the importance of the rainfall. I believe this would be a perfect addition to the final discussion provided in the conclusions, where the authors (already) examine some of the limits of their work.

*Response:  In all three experiments, we used only "simulation" setting, no "forecasting" setting.*

2. The other thing, which is probably clear in general, and most likely just a result of my oversight or bad memory is the following question: Why are the reported results in this version so much worse than in the first manuscript version?

*Response: As the reviewer's comment on the first revision, if we put the discharge in the input, the relationship between it and the output would affect the analysis of the effect of different types of rainfall on the results. In the new version, we removed the discharge from the input which is the reason for the discrepancies between the two results.*

*3-4: Response: Thanks to your suggestion, I have rewritten the LSTM description and added references.*

*5-8: Response: Thanks to your suggestion, I have rewritten the sentences based on your comments. I think the sentence ("The regional setting is of particular interest because it allows the model to encapsulate different hydrological processes by learning from more data and situations.") you suggested is very helpful and I have added it to the essay as well.*

*9: Response: Thanks to your suggestion. Considering that rainfall is the most direct and influential factor on rainfall-runoff simulation, the main objective of this study is to compare the difference between the results obtained using the LSTM model driven by rainfall data with spatial distribution information and the LSTM model driven by basin mean rainfall data. I added the description in the introduction. In addition, I also mentioned at the end of the paper that subsequent studies will consider combining more factors with spatial distribution information on this basis. There are some issues that need to be addressed if considering combining more factors, such as differences in resolution. This is also part of the follow-up study.*

*10: Response: I have rewritten the RNN description and added references.*

*11-12: **Response:** I put the explanation of all variables in front of the formulas.*

*13: **Response:** There are two main reasons for using two regional models. The first reason is that this allows comparing the results of different regional models. The second reason is that catchments located in the same area have similar regional characteristics and rainfall runoff relationships. A regional model that performs reasonably well across all catchments within a region could potentially be a step towards the simulation of runoff for such catchments.*

*14. **Response:** Thanks to your suggestion. We did not do a formal hyper-parameter search. I have added the relevant description in Experimental Setup.*

*15. **Response:** I modified the relevant descriptions and moved them to Experimental Setup. I think the point you raise is very relevant. In that study, we counted the average length of vectors characterizing spatial distribution information for 10 watersheds (excluding catchment 6), and then we tried three lengths of 20, 30 and 40, and the results showed that 20 gave the best results. In fact the research we are doing now is related to the length of the vector, that is, how long the vector is(how much information it may imply), and the relationship between the simulation results. The size of the catchment, the heterogeneity of the rainfall distribution, etc. all have a potential impact on this relationship. We did not add the average rainfall as a supplement because the average rainfall value would potentially provide additional information compared to zero. This information may affect the relationship between spatial distribution information and the results.*

16. Minor comments

***Response:** Thank you for these comments, we have modified the sentences and tables accordingly.*

**Authors' response for Anonymous Referee #2**

1. Please zoom in one (or two) pick flow and present the variation of simulated and observed flows along with the rainfall used in the simulation. Please choose a long look back window. It will be useful to see if there are any lags or not (which cannot be seen clearly in Fig 7). It is important to show that the model(s) built is sensitive to rainfall and does not suffer from the influence of long look back window(s).

*Response: Thank you for your suggestion. In the latest version of the manuscript, I have updated Figure 4, Figure 5, and Figure 7. In each figure, I show the corresponding runoff process, while I zoom in to show two of the time periods with rainfall information.*

2. Other comments

a-d
*Response: Thank you for your comment. I have modified the sentences and tables accordingly.*

e
*Response: Thank you for your comment. The ID of each watershed is shown in Table 1.*

f-g
*Response: Thank you for your comment. I have modified the sentences and tables accordingly.*

h
*Response: Thank you for your comment. D is defined in the text at the top of the table.*